# Improving Metabolic Syndrome in Ghanaian Adults with Type 2 Diabetes through a Home-Based Physical Activity Program: A Feasibility Randomised Controlled Trial

**DOI:** 10.3390/ijerph20085518

**Published:** 2023-04-14

**Authors:** Mohammed Amin, Debra Kerr, Yacoba Atiase, Misbah Muhammad Samir, Andrea Driscoll

**Affiliations:** 1Centre for Quality and Patient Safety, Institute for Health Transformation, Faculty of Health, School of Nursing and Midwifery, Deakin University, 221 Burwood Highway, Burwood, VIC 3125, Australia; 2University of Ghana School of Medicine and Dentistry, National Diabetes Management and Research Centre, Korle-Bu Teaching Hospital, Accra P.O. Box GP4236, Ghana; 3Physiotherapy Department, Korle-Bu Teaching Hospital, Accra P.O. Box GP4236, Ghana

**Keywords:** physical activity, exercise, type 2 diabetes, randomized controlled trial, metabolic syndrome, cardiovascular risk factors, feasibility study

## Abstract

There is a high prevalence of metabolic syndrome (MetS) among people with type 2 diabetes mellitus (T2DM). Physical activity has the potential to improve health outcomes for individuals with type 2 diabetes. Our study aim was to determine the effect of a 12-week culturally appropriate home-based physical activity program on metabolic syndrome markers and quality of life in Ghanaian adults with T2DM. A secondary objective was to examine the feasibility of implementing the PA program. A feasibility randomised controlled trial (RCT) was conducted. A purposive sample of 87 adults with T2DM at the Korle-Bu Teaching Hospital, Ghana, were randomized into either the control group (CG) (*n* = 43) or the intervention group (IG) (*n* = 44). Participants in the IG received the physical activity program in addition to their usual diabetes care; those in the CG received their usual diabetes care. Measurements for feasibility, MetS markers, and quality of life (SF-12) were performed at baseline and 12-week follow-up. Following the 12-week program, participants in the IG showed a significant improvement in fasting blood glucose (2.4% vs. 0.4%, *p* < 0.05), waist circumference (5.4% vs. 0.4%, *p* < 0.05), and systolic blood pressure (9.8% vs. 1.5%, *p* < 0.05). There were no statistical differences between the IG and CG regarding high-density lipoprotein, triglycerides, and diastolic blood pressure at the 12-week follow-up. Classification of MetS were reduced in the IG compared to the CG (51.2% vs. 83.3%, *p* < 0.05). The MetS severity score improved in the IG compared to the CG (8.8% vs. 0.5%, *p* < 0.05). The IG improved in two of the eight SF-12 dimensions (physical function and vitality, *p* < 0.05) compared to the CG. Thirty-two (72.7%) participants completed all 36 exercise sessions. Another 11 (25%) participants completed 80% of the exercise sessions. No adverse events were reported. In conclusion, a 12-week home-based physical activity program is feasible and safe. The intervention has the potential to improve MetS and quality of life in Ghanaian adults with T2DM. The preliminary findings of this study need to be confirmed in a large-scale multi-centre RCT.

## 1. Introduction

Metabolic syndrome (MetS) is a condition closely associated with type 2 diabetes (T2DM) [1]. It consists of multiple cardiovascular risk factors including high blood pressure (BP), triglycerides (TG), fasting blood glucose (FBG), waist circumference (WC), and a reduced high-density lipoprotein (HDL) [1]. Having MetS as a comorbidity in T2DM negatively impacts peoples’ quality of life (QoL), as well as increasing their risk of mortality [1,2].

Evidence suggests that about 90% of Ghanaians with T2DM have MetS [3]. The high prevalence may be due to healthcare disparities for individuals living in Ghana, culminating in late diagnosis when disease complications are already showing [4]. This might also explain why 80% of diabetes-related deaths occur in low–middle income countries [5].

Research has identified that physical activity (PA) improves MetS, reducing a person’s risk for future cardiovascular event and premature death [6]. Physical activity refers to any bodily movement that results in energy expenditure, while exercise refers to planned, structured, and repetitive physical activity aimed at improving or maintaining physical fitness and health [7]. There is strong evidence that PA in people with T2DM provides significant health benefits including increased cardiorespiratory fitness and physical function, improved body composition, reduced metabolic risk, and an improved quality of life [7]. According to the American College of Sports Medicine (ACSM), while regular resistance training in people with T2DM improves insulin sensitivity and muscle mass, aerobic exercises improve daily hyperglycemia [8]. In particular, patients derive greater benefit when they combine aerobic and resistance exercises, compared to either modality alone [8]. Furthermore, some evidence suggests that regular walking, a low cost, easy-to-access aerobic exercise, can improve glycemic control, blood pressure, lipid profile, body composition, and quality of life in people with T2DM [9].

Despite the purported benefits of PA, the specific component of PA needed to achieve these improvements remains unclear. Additionally, people with T2DM generally are also identified as physically inactive [10] This problem is profound in developing countries, including Ghana where the number of people with T2DM who undertake regular PA is barely 21.4% [11,12].

Context-specific evidence of the spectrum of T2DM is needed in prioritising and designing relevant health interventions to address the challenges associated with T2DM complications [4]. ‘One-size-fits-all’ programs are often implemented with minimal benefit [13]. Thus, any meaningful PA program must target specific personal, environmental, and socio-cultural factors which prevent individuals from engaging in exercise. In Ghana, sociocultural barriers to diabetes self-care exist, including the perception that diabetes is a spiritual curse [14]. Some also rely on alternative medicine including herbal medicine and the use of concoctions in diabetes treatment [15]. Currently, no culturally targeted PA program exist for Ghanaian adults with T2DM. In this study, we implemented a PA program which was designed in line with the characteristics of the Ghanaian population. The objective was to determine the effect of the PA program on MetS markers and QoL in Ghanaian adults with T2DM. A secondary objective was to examine the feasibility of implementing the program.

## 2. Materials and Methods

### 2.1. Study Design

This study is a single-site parallel feasibility randomised controlled trial (RCT). The study design has the potential to provide information on intervention compliance, retention, adherence, recruitment, and safety [16,17]. Thus, it informs future trials on study design, relevant resources, trial personnel, and data management [17]. Given that this study is part of a PhD with limited funding, a feasibility RCT is appropriate.

Ethics approvals were obtained in May 2022 from the Deakin University Human Research Ethics Committee (Project Number 2022-041) and Institutional Review Board of the Korle-Bu Teaching Hospital (protocol ID number: KBTH-IRB/00011/2022). The study protocol was also registered with the Australian New Zealand Clinical Trials Registry (registration number: ACTRN12622000323729p). The conduct and reporting of this trial align with the Consolidated Standards of Reporting Trials (CONSORT) guidelines. Individuals provided written consent to participate in this study.

There was no a priori sample size calculation as PA programs had not been undertaken previously in Ghana. This project was guided by evidence that feasibility RCTs have a median sample size of 72 (36 per arm) [18]. Hence, the study sample was 87 participants, involving 44 in the intervention group (IG) and 43 in the control group (CG). This sample size allowed adequate assessment of study outcomes and an anticipated attrition of approximately 20% [18].

### 2.2. Participants

Between June and August 2022, 134 patients with T2DM and MetS who routinely reported to the National Diabetes Management and Research Centre (NDMRC) of the Korle-Bu Teaching Hospital (KBTH) were screened for possible recruitment into the study. A diagnosis of T2DM was confirmed by an NDMRC nurse or physician based on documentation in participant’s medical record. Metabolic syndrome was determined according to the definition of The National Cholesterol Education Program (NCEP) Adult Treatment Panel (ATP) III which requires the presence of any three of the following five cardiovascular risk factors: abdominal obesity, raised triglycerides, low HDL, raised BP, high FBG [19].

Participants were included in this trial if they: (a) had a diagnosis of T2DM and met the criteria of MetS, (b) received medical clearance from their physician to take part in the PA program, (c) were sedentary (not achieving 600 metabolic equivalent of task—minutes/week, based on the International Physical Activity Questionnaire—short form (IPAQ-SF) assessment [20]), (d) were a resident in Ghana for at least 2 years, and (e) aged at least 18 years old. Patients were not eligible to participate in this study if they: (a) had any medical condition which did not allow them to engage in PA or compromised their safety (e.g., severe heart or respiratory disease, arthritis, etc.), (b) had a frailty index score greater than 3 [21], (c) could not travel to KBTH for weekly exercise sessions, and (d) did not have access to a phone for follow-up. All participants underwent electrocardiogram (ECG) examination before the physician issued medical clearance.

Participants were randomly allocated using a computer-generated excel program in a 1:1 ratio to either the IG or the CG. A researcher independent to the research program, who had no relationship with participants, conducted the randomisation. Recruitment and data collection were undertaken by blinded assessors—research assistants and a biomedical scientist. Participants and physical therapists were not blinded to group allocation due to the nature of the intervention.

### 2.3. Study Interventions

#### 2.3.1. Control Group (Usual Diabetes Care in Ghana)

Usual diabetes care in Ghana involves routine follow-up visits to the diabetes clinic. Depending on the patient’s glycaemic control, this visit may occur every 1 to 6 months. During these visits, the patient undergoes routine medical assessment (e.g., weight, BP, urine analysis, and blood tests such as Hb1Ac, FBG, and lipid profile). Patients may also be referred for dietary management, and eye or foot care. Nurses also deliver a 30–45-min group education session to all clients of the clinic on self-care practices including exercise, medication adherence, dietary modification, foot care, and the need for follow-up visits. There is no structured PA program as part of the usual diabetes care. However, group exercise advice is delivered by general nurses to patients during clinic wait times.

#### 2.3.2. Intervention Group (PA Program)

Participants engaged in a structured 12-week PA program in addition to their usual diabetes care. In total, participants engaged in 36 exercise sessions, involving three sessions per week with no more than two days without exercise. With each session lasting 45 min, participants performed three minutes of stretching exercise, 10 min of resistance training using a resistance band, 30 min of walking (brisk pace, 5.2–5.6 km/h), and 2 min of cool-down exercises. Participants attended a weekly individual session at the physiotherapy department of the KBTH, under the supervision of a physical therapist (PT). All other sessions were self-delivered at home. Participants attended a 45-min group exercise education session prior to exercise sessions. The PT gave advice on the PA program, including its importance in controlling diabetes. Participants maintained an exercise diary and a PT followed individuals on a weekly basis to encourage and document exercise participation. All participants in the IG received a brochure, a resistance band, and videos that displayed the prescribed exercise. Figure 1 shows a timeline of the intervention program.

### 2.4. Outcome Measures

The primary outcome measure was a change in MetS markers (BP, TG, WC, FBG, and HDL) in the IG, compared to those in the CG. Secondary outcomes included: (1) a change in QoL in the IG, compared to the CG; and (2) feasibility of implementing the PA program determined by measures of participation and safety. Except for measures of participation and safety which were assessed each week, the primary and other secondary outcomes were assessed at baseline and 12-week follow-up.

A demographic data questionnaire was used to collect personal information including age, sex, and marital, educational and employment status. For pathology measures, participants fasted for 8 h overnight, and a blood sample (5 mL) was collected by a biomedical scientist for the following measures: TG, HDL, and FBG. Waist circumstance was measured by one nurse, independent of the research team, measuring the mid-point between the lower costal margin and iliac crest. Blood pressure measurement was taken after the individual had rested for 10 min, in a seated position. Three consecutive readings with 5-min intervals were collected using a calibrated sphygmomanometer, and the average of the last two readings were used for analysis.

Guided by the formula by Johnson et al. [22], the MetS Z-score was calculated and considered as a continuous variable for the five MetS markers (FBG, WC, TG, BP, HDL), in line with the NCEP ATP III guidelines. The MetS Z-score is a specific measure of MetS severity, and it is used to predict a person’s total cardiometabolic risk status. The following formula by Johnson et al. [22] was used to calculate the MetS Z-score: men, [(40 − HDL)/9.0] + [(TG − 150)/81.0] + [(GLU − 100)/11.3] + [(WC − 102)/7.7] + [(MAB − 100)/9.1] versus [(50 − HDL)/14.1] + [(TG − 150)/81.0] + [(GLU − 100)/11.3] + [(WC − 88)/9.0] + [(MAB − 100)/9.1] for women.

Quality of life was assessed using the Short Form Health Survey—version 2 (SF-12V2). The 12-item tool was used to measure eight specific health domains: physical functioning, role limitation related to physical functioning, role-physical, bodily pain, general health, vitality, social functioning, role-emotional, and mental health. The items on the scale were scored from 0 to 100, where 100 indicates high quality of life [23].

Measures of participation were assessed using the exercise diary, by recording the proportion of exercise sessions that participants completed. The exercise diary was developed by the authors, and its face validity was verified by the research team which included an endocrinologist, nurse, and PT. The diary was used to document the type, length, intensity, and frequency of exercise performed by participants each week. During the weekly visit to the physiotherapy department at KBTH, the PT asked participants about the exercise they performed during the week. A participation rate of more than 70% of the scheduled sessions (36 in total) was considered as high participation. Reasons for missing a session were also documented. Measures of safety were determined by participants’ subjective reports of any adverse events from the exercise sessions.

### 2.5. Data Analysis

Data were analysed using STATA (version 17). Continuous data are reported as means +/− SD and categorical data are reported using frequencies and percentages. Baseline comparisons between the control group and the intervention group were analysed using two sample t-tests or chi-squared tests for continuous and categorical variables, respectively. A Fisher’s exact test was used if there were small samples (i.e., frequencies less than 5). To compare the groups in this study, statistical analyses were performed with linear mixed models. Linear mixed models were used because in all methods, an adjustment should be made for the dependency of the repeated observations within the individual (i.e., at baseline and at follow-up). The effect size was calculated using Cohen’s d, which is the follow-up mean minus the baseline mean, divided by the pooled standard deviation. In all instances, a *p*-value < 0.05 was considered statistically significant.

## 3. Results

### 3.1. Sample Characteristics

A total of 87 out of 134 patients screened for eligibility were included in the study, indicating a recruitment rate of 64.9%. Reasons for exclusion included poor ECG results (*n* = 12), severe neuropathy (*n* = 10), unable to travel to the hospital for weekly exercise session (*n* = 7), not having a mobile phone (*n* = 4), and declined to participate (*n* = 14). Some individuals had multiple exclusion criteria.

Forty-four patients were randomly allocated to the IG (mean age 56.2 ± 8.3 years) and 43 were assigned to the CG (mean age 56.5 ± 8.5 years). More females participated in the study (65.5%). On average, patients had been diagnosed with T2DM for 6.1 ± 2.7 years. Most participants were taking oral glycaemic agents (89.40%); an additional 11.4% participants were prescribed insulin. All participants met the diagnostic criteria for MetS. There were no significant differences between the two groups for baseline characteristics, as shown in Table 1. Medication regimes remained unchanged throughout the trial for all participants. There were no medication variations at baseline for each of the groups.

### 3.2. Physical Activity Program Participation

Thirty-two (72.7%) participants in the IG completed the 12-week program (all 36 sessions at home and facility) and the remaining eleven (25%) participants completed 80% of the exercise sessions. The average number of sessions completed was 29 ± 5.4 (mean ± SD) out of 36 sessions. This indicates high feasibility. Reasons for missing sessions included high work demands, social commitments, not feeling well, and long travel hours for the weekly facility exercise sessions.

Eight participants (IG = 1, CG = 7) did not complete the program. One participant from the IG withdrew from the study due to hospital admission for medical reasons not related to the PA program. Seven participants in the CG were lost to follow-up due to: travel away from home (*n* = 2), loss of interest (*n* = 2), and being unable to be contacted (*n* = 3). A CONSORT flow diagram of the trial and schedule of visits is presented in Figure 2.

### 3.3. Effect of Physical Activity on Metabolic Syndrome Markers

Following the 12-week PA program, participants in the IG had significant improvement (*p* < 0.05) in WC, SBP, and FBG compared to participants in the CG, as shown in Table 2. There were no significant improvements for TG, DBP, and HDL (*p* > 0.05) between the two groups. Further, there was greater improvement in the MetS severity score for the IG (8.8% change) compared with the CG (0.5% change). At enrolment, all participants met the criteria for MetS. After the 12-week study period, 51.2% in the IG and 83.3% in the CG met criteria for MetS, a statistically significant difference (*p* < 0.05).

Correlation analyses showed no significant difference between groups for oral glycaemic medication use and insulin use (*p* > 0.05).

### 3.4. Effect of Physical Activity on Quality of Life

Comparing the IG to the CG, our results show that QoL generally improved in some dimensions of the SF-12, in favour of the IG, as shown in Table 3. Participants in the IG had a significant improvement in the physical component summary score (*p* = 0.000) compared to the CG. The IG showed significantly higher scores compared to the CG in two dimensions of QoL, following the 12-week intervention program: vitality (*p* = 0.008) and physical functioning (*p* = 0.000). No significant improvements were observed in the other dimensions: role limitations—physical health, bodily pain, general health, social functioning, role limitations—emotional, and mental health.

## 4. Discussion

The findings of this study show that the culturally appropriate 12-week PA program designed specifically for Ghanaian adults with T2DM improved three MetS markers (FBG, SBP, and WC). This is likely to have contributed to the reclassification of 48.8% of participants in the IG as no longer having MetS after completion of the PA program. Further, there was high engagement in the program. Except for one participant who dropped out of the program, all participants in the IG completed at least 80% of the exercise sessions, with no exercise-related adverse events recorded during the study period.

The exercise program trialled in this study involved a combination of aerobic and resistance exercises, with our study results showing a positive impact on MetS severity score among the IG compared to the CG. This finding is consistent with guidelines from the American College of Sports Medicine, suggesting that combined exercise in people with T2DM produced more beneficial outcome compared to aerobics or resistance exercise alone [10]. There is evidence that aerobic exercise is the most beneficial mode of exercise for individuals with MetS without T2DM [24]. In addition, Pattyn et al. [25] conducted a meta-analysis among a similar population and found that endurance training has a favourable effect on cardiovascular risk factors that are a component of MetS. The findings from the two studies identified the lack of data regarding the effect of combined training on MetS markers in people with T2DM. The current study, which involved combined exercise (aerobics and resistance), contributes in filling this gap.

The overall improvement in MetS severity score following exercise intervention cannot be overemphasised. It is recommended that studies investigating the role of exercise in reducing MetS markers should highlight whether participants still meet the MetS criteria after receiving an intervention [24]. Our current study shows that the improvement in MetS markers resulted in reclassification of 48.8% of participants in the IG as no longer having MetS. A previous study explored the efficacy of exercise training in treating MetS in a sample of 621 sedentary and healthy individuals enrolled in a 20-week exercise program. At the end of the program, 30.5% of the participants were no longer classified as having MetS [26]. Relative to the findings of this current study, there is a comparable outcome between healthy adults and people with T2DM, in terms of reduction in MetS severity score following a PA intervention program.

Wewege et al. [24] undertook a meta-analysis to examine the impact of exercise in reducing MetS markers in healthy adults. Eleven studies were included. The results showed a significant improvement in WC (3.4 cm), FBG (0.15 mmol/L), HDL (0.05 mmol/L), TG (0.29 mmol/L), and DBP (1.6 mmHg). Similarly, we found in our current study that individuals who participated in the PA program were more likely to have a reduced WC, SBP, and FBG, which could potentially reduce cardiovascular risk [27].

Our analysis of changes in QoL following the PA intervention program revealed significant improvement in two dimensions: physical function and vitality. Few studies have examined the effect of PA on QoL in adults with T2DM, and there has been inconsistency in their findings. Using SF-36 to examine the impact of resistance training on the mental health status of Puerto Rico adults with T2DM, Lincoln et al. [28] found positive improvement in QoL following a 16-week PA program. In contrast, Plotnikoff et al. [29] did not find any significant improvement for the mental and physical composite scales of QoL for adults with T2DM who participated in a 16-week PA program. Another study utilising exercise bands in a 4-month intervention program for people with T2DM did not show significant improvement in functional capacity [30]. Previous study examining the impact of a 16-week resistance exercise on the mental health of adults with T2DM found positive effects of high-intensity progressive resistance exercise training on mental health status [28]. Similarly, there is some evidence that an 8-week supervised program involving resistance and aerobic exercises improved some aspect of QoL, such as general health in adults with T2DM [31]. The varied reports may be attributed to the limited number of studies, varied QoL assessment tools, and diverse designs of the PA program.

Our study had a high participation rate. Previous studies in Ghana have reported low PA participation (21.4%) among the Ghanaian population living with T2DM [12]. Low participation rates in similar populations in Malawi, Botswana, and Sri Lanka have identified barriers to exercise such as lack of knowledge and social support, lack of exercise facilities [32], fear of social ridicule, and functional limitations including poor health [33,34]. The high participation and retention rate in this study is comparable to participation rates in previous studies conducted in Sri Lanka—82% [35], India—97% [36], and Canada—71% [29]. The high participation rate in this study could possibly be attributed to the structured nature of the program that included a home-based component which may be suitable for this population. There is some evidence that certain communities in Ghana do not favour women publicly engaging in exercise because of the belief that women should not expose their body shapes, as expressed in exercise attire [37]. Additionally, the lack of exercise facilities including community parks and proper walkways in Ghanaian communities is cited as a major barrier to exercise uptake among the general population [38]. Therefore, a self-delivered low-cost home-based PA program might have been feasible for participants in this study, which probably accounted for the high retention and participation rate. Furthermore, the education component of the PA program implemented in this study may have contributed to increasing participants’ awareness and motivation.

A major strength of this study is that there were no significant differences between the two groups at baseline. Moreover, participants received weekly follow-up with a PT. In the first week, all exercise sessions were supervised by a PT at the physiotherapy department to ensure that exercises were performed correctly and within safety limits. Furthermore, the study was conducted in the largest diabetes clinic in Ghana, where patients are from diverse socio-economic and cultural backgrounds.

A limitation of this study is the subjective report of participants’ weekly PA, which could possibly lead to under or over reporting. There is some evidence of a fair agreement between self-reported and objectively measured PA, despite evidence that about 59% of patients over-report their PA levels [39]. To encourage participants’ retention in this current study, 22 of the 36 exercise sessions were self-delivered at home. Participants were asked to document and report their participation in exercise to the PT during the weekly session at the hospital. The PT made follow-up calls and sent text messages each week to remind participants to do their home exercises. Another limitation of the study was that individuals in the CG may have increased their exercise levels after receiving the routine lifestyle modification education session at the diabetes clinic. Prior to enrolling participants in the CG, they were counselled to continue their usual diabetes care.

People who volunteer in PA programs may have high self-motivation; therefore, this might have accounted for the high participation rate. A plausible reason for the successful recruitment and retention of participants may also be related to the tests performed as part of enrolment in this study. (e.g., pathology tests, ECG). The costs for these tests were covered by the funds available for this research, which is self-funded. These expenses pose a financial burden for patients, hence may have contributed to their decision to participate. Lastly, the sample size was small, and it was a single-site study. However, a feasibility study was conducted within the constraints of limited funding and PhD timelines.

## 5. Conclusions

Overall, our findings suggest that the PA program implemented in this study has the potential to improve MetS and QoL in Ghanaian adults with T2DM. Further, the PA program was feasible and safe. With a signification proportion of Ghanaians with T2DM resorting to alternative diabetes treatment, coupled with barriers to diabetes self-care, HCPs in Ghana have a responsibility to promote positive lifestyle behaviours. Further multi-centre studies involving a larger sample size are warranted.

## Figures and Tables

**Figure 1 ijerph-20-05518-f001:**
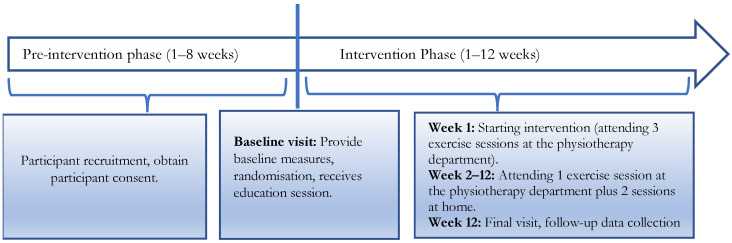
Intervention timeline.

**Figure 2 ijerph-20-05518-f002:**
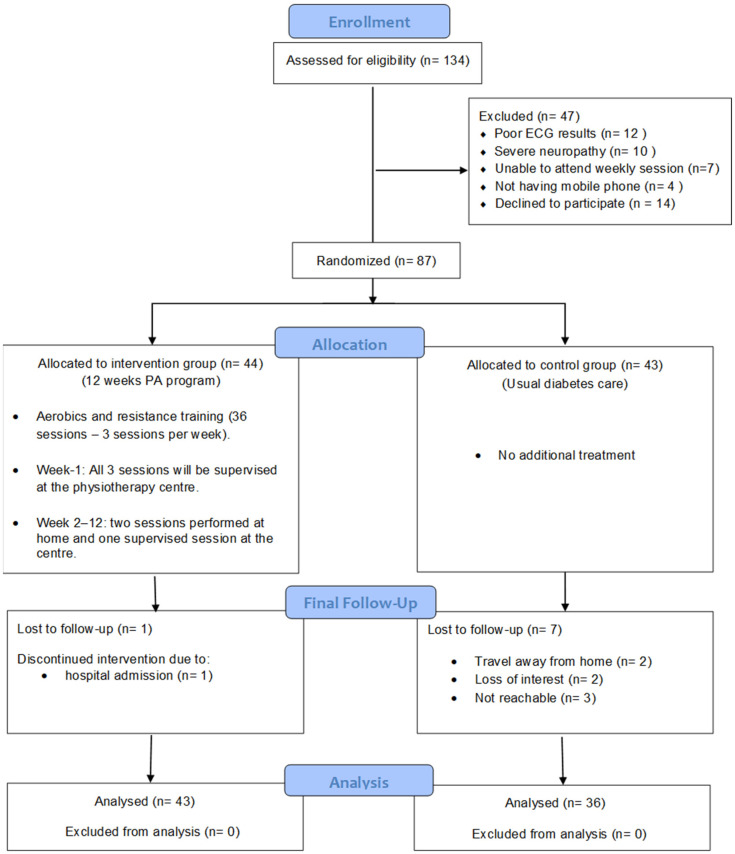
Participant flow chart.

**Table 1 ijerph-20-05518-t001:** Baseline characteristics of participants.

Characteristics	Intervention Group	Control Group	*p*-Value
	(*n* = 44)	(*n* = 43)	
Age	56.2 ± 8.3	56.5 ± 8.5	0.89
Gender (%)			0.41
Male	38.6	30.2	
Female	61.4	69.8	
Education status (%)			0.96
No formal/primary	77.3	79.1	
Secondary	15.9	14.0	
Tertiary	6.8	6.9	
Not employed (%)	63.6	62.8	0.93
Married (%)	34.1	30.2	0.70
Dietary management (%)	6.8	2.3	0.31
Insulin treatment (%)	11.4	11.6	0.61
Oral glycemic agents (%)	86.4	93.0	0.25
Antihypertensive agents (%)	90.7	91.0	0.97
Lipid-lowering agents (%)	68.2	79.1	0.25
MetS markers (%)			
High FBG	100	97.7	0.31
Reduced HDL	18.2	25.6	0.40
High triglycerides	34.1	28.0	0.53
High BP	70.5	65.1	0.60
High WC	95.5	100	0.16
Duration of diabetes	6.1 ± 2.7	6.7 ± 3.3	0.33
PA level (MET)	384.6 ± 45.8	387.6 ± 40.6	0.74

*p*-Value is <0.05, mean ± SD, %, MET = metabolic equivalent of task, FBG = fasting blood glucose, HDL = high density lipoprotein, BP = blood pressure, WC = waist circumference.

**Table 2 ijerph-20-05518-t002:** Effect of PA on metabolic syndrome markers.

Variable	Intervention Group				Control Group				
	Before *n* = 44	After *n* = 43	Δ %	ES	Before *n* = 43	After *n* = 36	Δ %	ES	*p*-Value
WC (cm)	99.1 ± 12.9	93.7 ± 11.9	5.4	0.43	99.4 ± 11.1	99.8 ± 11.4	0.4	0.03	0.02 *
HDL (mg/dL)	1.4 ± 0.4	1.8 ± 0.5	0.4	0.25	1.6 ± 0.4	1.7 ± 0.5	0.1	0.77	0.11
TG (mg/dL)	1.4 ± 0.6	1.6 ± 0.4	0.2	0.52	1.3 ± 0.5	1.4 ± 0.5	0.1	0.28	0.37
SBP (mmHg)	141.5 ± 17.7	131.7 ± 14.3	9.8	0.61	135.6 ± 17.9	137.1 ± 14.9	1.5	0.09	0.05 *
DBP (mmHg)	79.4 ± 9.7	78.6 ± 9.4	0.8	0.08	78.3 ± 12.3	78.2 ± 9.8	0.1	0.01	0.41
FBG (mmol/L)	8.3 ± 1.9	5.9 ± 0.6	2.4	0.22	8.3 ± 2.6	7.9 ± 1.8	0.4	1.66	0.01 *
MetS Z-score	211.2 ± 23.6	202.4 ± 16.4	8.8	0.43	211.3 ± 14.2	211.8 ± 14.9	0.5	0.03	0.001 *

Mean ± SD, WC = waist circumference, HDL = high density lipoprotein, TG = triglycerides, SBP = systolic blood pressure, DBP = diastolic blood pressure, FBG = fasting blood glucose, MetS Z-score = metabolic syndrome severity score, ES = effect size, Δ % denotes percentage change, * denotes statistical significance, *p*-value is <0.05.

**Table 3 ijerph-20-05518-t003:** Effect of physical activity on quality of life.

	Intervention Group			Control Group			
	Before	After	ES	Before	After	ES	*p*-Value
	*n* = 44	*n* = 43		*n* = 44	*n* = 36		
Physical functioning	67.4 ± 15.6	87.5 ± 7.6	1.63	64.7 ± 9.9	67.8 ± 8.4	0.34	0.000 *
Role—physical	72.4 ± 10.6	78.4 ± 9.0	0.60	60.7 ± 10.0	63.8 ± 8.4	0.33	0.343
Bodily pain	62.1 ± 10.2	59.1 ± 11.0	0.29	55.9 ± 9.6	56.1 ± 10.9	0.01	0.318
General health	72.1 ± 9.6	69.4 ± 11.0	0.25	66.7 ± 8.3	66.9 ± 11.6	0.21	0.359
Vitality	70.3 ± 7.7	75.8 ± 8.9	0.66	72.0 ± 7.9	71.3 ± 7.6	0.09	0.008 *
Social functioning	55.11 ± 10.2	54.2 ± 11.1	0.09	49.2 ± 9.8	49.1 ± 10.9	0.01	0.787
Role—emotional	67.1 ± 10.1	69.0 ± 11.6	0.18	60.9 ± 9.6	61.1 ± 11.6	0.12	0.577
Mental health	54.9 ± 8.3	58.1 ± 10.2	0.29	55.0 ± 7.4	55.1 ± 9.6	0.21	0.138
Physical component summary	56.4 ± 15.6	76.5 ± 7.6	1.63	53.7 ± 9.9	56.8 ± 8.4	0.34	0.000 *
Mental component summary	56.4 ± 15.6	56.4 ± 15.6	0.01	56.4 ± 15.6	56.4 ± 15.6	0.29	0.318

SF-12 range 0–100, where 100 = high QoL paired-sample *t*-test, * indicates significant changes between baseline and 12-week follow-up; significant at 0.05 level.

## Data Availability

The data presented in this study are available on request from the corresponding author.

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
