# Peer review of "Improving Metabolic Syndrome in Ghanaian Adults with Type 2 Diabetes through a Home-Based Physical Activity Program: A Feasibility Randomised Controlled Trial"

_ijerph, 2023, doi:10.3390/ijerph20085518_

Round 1
Reviewer 1 Report
In this manuscript the authors explore the effects of physical activity on patients with type 2 diabetes and metabolic syndrome. The text is well structured, although there are some issues that need to be addressed. The use of English language is adequate, I made some minor suggestions for improvement.
My main issues are listed below
Line 91 please expand IG and CG, as this is its first mention
Lines 77 to 93 belong to material and methods section, and are in it word for word, please fix this
Line 117 a nurse can make a diagnosis of T2DM? Is this appropriate in Ghana?
Line 121 and further please make inclusion and exclusion criteria clearer
Line 154 I would suggest not to use trade mark names like TheraBand, just say resistance band
Line 157 please expand PT as it has not been mentioned earlier
Line 177 something is missing before the reference number
Line 251 tables are mislabeled, and need to be moved closer to their first mention in the text, this also applies to figures
Figure 1 as it is written now it appears that CG received no treatment, but as it is stated in the text they received their usual therapy for DM, also fix the final follow up in CG there is “n=” missing
Table 3 is not mentioned in the text.
Line 308 please change potentially improve cardiovascular risk to decrease (or reduce) cardiovascular risk, if this better describes what you wanted to state
Line 324 please choose another word instead of costume like attire or another...
Some of the information about PT activity like remaining patients to exercise are given in discussion, and not in material and methods section, I would suggest to mention all of them in M&M section
At few instances in the text you refer to your own papers that are still in review process, I would discourage this because you do not know if and in which form they will be published
I do not understand the need for anonymization of the text because it is clear where the study was conducted and it is clearly mentioned in the acknowledgments section.
In the text you mention that the costs were paid for by research funds (line 370), and in the funding section (line 389) you state that you received no funding?
Author Response
Thank you for your comments. Your comments have been considered in the new draft.

Reviewer 2 Report
This is a very interesting study which aims to improve the quality of life in Type 2 diabetic patients.
My suggestion to the authors is that they improve their results section, perhaps include graphs instead of tables only. Especially table 2, can be presented in a graph format.
Author Response
Thank you for your comments which has guided this new draft

Reviewer 3 Report
In this manuscript, Mohammed Amin and other authors demonstrated that the 12-week PA program designed specifically for XXXian adults with T2DM improved three MetS markers (FBG, SBP, and WC). From the statistic results, participants following the PA program had significant improvement in some of the metabolic syndrome markers compared with participants in the control group. But I have two questions about the experiment design.
First, does the PA program have the same effect on healthy people without T2DM and MetS?
Second, what is the difference between MetS patients and healthy persons before and after taking the PA program for 12-week?
Besides, in line 251 the table title should be Table 1 instead of Table 2.
Author Response

(The authors gave the same response as above.)

Reviewer 4 Report
The present paper offers an interesting insight of the development of a PA program in T2DM patients, especially concerning the home-based approach. Anyway, here are some suggestions I would recommend:
- You could expand the introduction and background paragraph including some detailed information concerning PA effects in T2DM patients (i.e. differences between aerobic and resistance training/distinguishing between physical exercise and PA such as walking)
- Lines 77-93 and 96-112 are repeated
- It would be clearer to have the participant flow chart after paragraph 3.2
- You specified the medication regimes remained unchanged throughout the trial. However, can you specify: if there were any medication variations at baseline for each group/if it is true for both glycemic agents and for other agents (e.i. hypolipidemic, antihypertensive drugs)?
- In the baseline participants characteristics, please include how many were taking hypolipidemic and antihypertensive drugs, and if there were any significant differences between groups
- The "baseline characteristic of participants" would be Table 1, not Table 2, please change it. Moreover, it would be appreciated if you could add to this Table also all the MetS markers. Also, please add the "%" to all the categorical variables in the Table (i.e. married or dietary management)
- Line 244: TG is repeated, please change one to HDL
- You could provide a timeline to better describe the program
Author Response

(The authors gave the same response as above.)

Round 2
Reviewer 1 Report
I do not have any significant suggestions.